# *Drosophila* macrophages switch to aerobic glycolysis to mount effective antibacterial defense

**Gabriela Krejčová[1]\*, Adéla Danielová[1], Pavla Nedbalová[1], Michalina Kazek[1], Lukáš Strych[1], Geetanjali Chawla[2†], Jason M Tennessen[2], Jaroslava Lieskovská[3,4], Marek Jindra[1,5], Tomáš Doležal[1], Adam Bajgar[1,5]\***

[1]Department of Molecular Biology and Genetics, University of South Bohemia, Ceske Budejovice, Czech Republic; [2]Department of Biology, Indiana University, Bloomington, United States; [3]Department of Medical Biology, University of South Bohemia, Ceske Budejovice, Czech Republic; [4]Institute of Parasitology, Biology Centre CAS, Ceske Budejovice, Czech Republic; [5]Institute of Entomology, Biology Centre CAS, Ceske Budejovice, Czech Republic

**\*For correspondence:**
krejcovagabriela@seznam.cz (GK);
bajgaa00@prf.jcu.cz (AB)

**Present address:** [†]Regional Centre for Biotechnology, Faridabad, India

**Competing interests:** The authors declare that no competing interests exist.

**Abstract** Macrophage-mediated phagocytosis and cytokine production represent the front lines of resistance to bacterial invaders. A key feature of this pro-inflammatory response in mammals is the complex remodeling of cellular metabolism towards aerobic glycolysis. Although the function of bactericidal macrophages is highly conserved, the metabolic remodeling of insect macrophages remains poorly understood. Here, we used adults of the fruit fly *Drosophila melanogaster* to investigate the metabolic changes that occur in macrophages during the acute and resolution phases of *Streptococcus*-induced sepsis. Our studies revealed that orthologs of Hypoxia inducible factor $1\alpha$ (HIF$1\alpha$) and Lactate dehydrogenase (LDH) are required for macrophage activation, their bactericidal function, and resistance to infection, thus documenting the conservation of this cellular response between insects and mammals. Further, we show that macrophages employing aerobic glycolysis induce changes in systemic metabolism that are necessary to meet the biosynthetic and energetic demands of their function and resistance to bacterial infection.
DOI: https://doi.org/10.7554/eLife.50414.001

## Introduction

Macrophages represent a highly specialized and versatile population of cells that occur in all animals and perform a diversity of functions (*Lim et al., 2017*). In the absence of an activating stimulus, macrophages reside as quiescent sentinel cells that have minimal metabolic requirements (*Davies and Taylor, 2015*). In response to extracellular triggers, however, macrophages undergo a dramatic change in behavior that coincides with an enhanced metabolic rate and increased energy demands (*Pearce and Pearce, 2013*). In this regard, the manner by which macrophages mount a response is dictated by the activating stimuli, which include tissue-damage-, pathogen- or microbe-associated molecular patterns (DAMPs, PAMPs and MAMPs, respectively), as well as signaling molecules that are secreted by other cells, such as cytokines. Each challenge requires the induction of specific metabolic and physiological processes that allow for an adequate immune response (*Kawai and Akira, 2011*) – cellular changes that are collectively known as a polarization phenotype.

Macrophages polarize into bactericidal (M1) or healing (M2) functional phenotypes characterized mainly by metabolism (*Mills et al., 2000*). M1 and M2 polarization phenotypes utilize distinct ways of generating ATP (glycolysis vs. oxidative phosphorylation) and metabolizing arginine (NO synthesis vs. the ornithine cycle) (*O'Neill and Pearce, 2016*). Nowadays, the whole spectrum of polarization

**eLife digest** Macrophages are the immune system's first line of defense against infection. These immune cells can be found in all tissues and organs, watching for signs of disease-causing agents and targeting them for destruction. Maintaining macrophages costs energy, so to minimize waste, these cells spend most of their lives in 'low power mode'. When macrophages sense harmful bacteria, they rapidly awaken and trigger a series of immune events that protect the body from infection. However, to perform these protective tasks macrophages need a sudden surge in energy.

In mammals, activated macrophages get their energy from aerobic glycolysis – a series of chemical reactions normally reserved for low oxygen environments. Switching on this metabolic process requires a protein called hypoxia inducible factor 1α (HIF-1 α), which switches on the genes that macrophages need to generate energy as quickly as possible. Macrophages then maintain their energy supply by sending out chemical signals which divert glucose away from the rest of the body.

Fruit flies are regularly used as a model system for studying human disease, as the mechanisms they use to defend themselves from infections are similar to human immune cells. However, it remains unclear whether their macrophages undergo the same metabolic changes during an infection.

To address this question, Krejčová et al. isolated macrophages from fruit flies that had been infected with bacteria. Experiments studying the metabolism of these cells revealed that, just like human macrophages, they responded to bacteria by taking in more glucose and generating energy via aerobic glycolysis. The macrophages of these flies were also found to draw in energy from the rest of the body by raising blood sugar levels and depleting stores of glucose. Similar to human macrophages, these metabolic changes depended on HIF1α, and flies without this protein were unable to secure the level of energy needed to effectively fight off the bacteria.

These findings suggest that this metabolic switch to aerobic glycolysis is a conserved mechanism that both insects and mammals use to fight off infections. This means in the future fruit flies could be used as a model organism for studying diseases associated with macrophage mis-activation, such as chronic inflammation and autoimmune diseases.

DOI: https://doi.org/10.7554/eLife.50414.002

phenotypes corresponding to particular functions has been described (*Mosser and Edwards, 2008*; *Martinez and Gordon, 2014*). Perhaps the most dramatic change in macrophage metabolism associates with the M1 bactericidal phenotype, in which cells increase both glucose consumption and lactate production independently of oxygen concentration - a phenomenon known as aerobic glycolysis (AG) (*Warburg et al., 1927*; *Warburg, 1956*). The resulting metabolic program promotes increased glucose catabolism, thus allowing M1 macrophages to generate enough of the ATP and glycolytic intermediates necessary for elevated phagocytic cell activity (*Liberti and Locasale, 2016*). This shift in cellular metabolism towards AG appears to be a determining factor in macrophage function and in the development of the pro-inflammatory phenotype (*Galván-Peña and O'Neill, 2014*).

Hypoxia inducible factor 1α (HIF1α) is a key regulator of AG within macrophages. Although this transcription factor is normally degraded in the presence of oxygen, the triggering of either Toll-like receptor (TLR) or Tumor necrosis factor receptor (TNFR) signaling within macrophages activates Nuclear factor kappa-B (NFKB) and stabilizes HIF1α independently of oxygen availability (*Siegert et al., 2015*; *Jung et al., 2003*). This normoxic HIF1α stabilization promotes the expression of genes that are under the control of hypoxia response elements (HREs), many of which are involved in cellular metabolism, cell survival, proliferation, and cytokine signaling (*Dengler et al., 2014*). In this regard, two of the key HIF1α target genes encode the enzymes pyruvate dehydrogenase kinase (PDK) and lactate dehydrogenase (LDH), which together shunt pyruvate away from the mitochondria and maintain $NAD^+$/NADH redox balance independently of oxidative phosphorylation. Inhibition of both HIF1α and LDH represents an efficient experimental strategy to direct cellular metabolism from AG to oxidative phosphorylation in both mice and *Drosophila* (*Allison et al., 2014*; *Geeraerts et al., 2017*), demonstrating the crucial role of these enzymes in this metabolic switch.

Although pyruvate metabolism within the tricarboxylic acid (TCA) cycle is limited during AG, the TCA intermediates are essential for many cellular processes. Therefore, cells under AG rely on

feeding the TCA cycle with glutamine, causing a TCA cycle to be 'broken' (*Langston et al., 2017*). Such a dramatic change in mitochondrial metabolism leads to significant imbalances in the cytosolic accumulation of TCA metabolites (such as NO, succinate, fumarate, L-2-hydroxyglutarate) that further contribute to HIF1α stabilization (*Bailey and Nathan, 2018*). While this feedback maintains AG, it simultaneously makes it dependent on a sufficient supply of nutrients from the environment (*Iommarini et al., 2017*).

Macrophages employing AG must consume sufficient carbohydrates to support biosynthesis and growth. In order to ensure an adequate supply of sugar and other nutrients, these cells produce signaling molecules that affect systemic metabolism in order to secure enough energy for themselves – a concept recently defined as selfish immune theory (*Jeong et al., 2003*; *Straub, 2014*). According to this theory, signaling molecules released by immune cells induce systemic metabolic changes such as hyperglycemia and systemic insulin resistance to increase the titer of nutrients that are available for the immune response and to limit their consumption by other tissues and processes (*Dolezal, 2015*). As many of these signaling molecules are direct HIF1α transcriptional targets, HIF1α stabilization directs the cellular metabolism while it simultaneously induces the expression of genes that have an impact on the whole systemic metabolism (*Peyssonnaux et al., 2007*; *Imtiyaz and Simon, 2010*). Thus, macrophages not only phagocytose cells but they also regulate the systemic metabolism of an organism.

*Drosophila* macrophages, like those of mammals, serve an essential role in the immune system and are capable of responding to a wide array of stimuli, ranging from pathogenic bacteria and fungi to the corpses of apoptotic cells (*Wood and Martin, 2017*; *Sears et al., 2003*; *Govind, 2008*). The mechanism of the bactericidal function itself is highly conserved at the molecular level between *Drosophila* and mammalian macrophages. In both *Drosophila* and mammals, two central signaling pathways, Toll and Imd (TLR and TNFR functional homologs) are triggered in response to pathogenic stimuli (*Valanne et al., 2011*; *Buchon et al., 2014*; *Lemaitre et al., 1996*). The Toll and Imd pathways induce the NFKB signaling in *Drosophila*, so we can assume that the phagocytic role of macrophages could be accompanied by stabilization of the HIF1α ortholog, *Similar* (*Sima*), hereafter referred to as Hif1α (*van Uden et al., 2011*). Indeed, normoxic stabilization of Hif1α followed by its nuclear localization and increased expression of HRE-controlled genes can induce metabolic changes that are typical of AG (*Romero et al., 2008*; *Li et al., 2013*; *Liu et al., 2006*; *Herranz and Cohen, 2017*; *Eichenlaub et al., 2018*). Even though the HRE-controlled genes frequently appear in transcriptomic data for activated insect macrophages (*Irving et al., 2005*; *Johansson et al., 2005*), the direct role of Hif1α in the macrophages has not yet been tested.

Considering that the molecular mechanisms that control macrophage activation are similar in both *Drosophila* and humans, it seems logical that the metabolic changes that occur within these cells would also be comparable, but the metabolism of insect macrophages remains poorly understood. Here, we address this question by analyzing in vivo metabolic and transcriptional changes in adult *Drosophila* phagocytic macrophages by employing a model of *Streptococcus-pneumoniae*-induced sepsis. The well-defined progress of this infection allowed us to distinguish three phases of the immune response according to the changing dynamics of bacterial growth (acute, plateau, and resolution phase) (*Figure 1A*). The acute phase lasts for the first 24 hr, during which the streptococcal population is rapidly growing and its abundance must be limited by phagocytosis to avert early death (*Pham et al., 2007*; *Bajgar and Dolezal, 2018*). The established equilibrium between continuous bacterial growth and host bacterial killing results in the plateau phase lasting for the next four days. At the end of this period, the immune system of the host surmounts the infection and clears the majority of the pathogens. The following resolution phase (120 hr post-infection (hpi) and later) is essential for macrophage-mediated clearance of bacterial residues and for the reestablishment of homeostasis (*Bajgar and Dolezal, 2018*; *Chambers et al., 2012*).

To analyze processes that are characteristic of highly active phagocytic macrophages in *Drosophila*, we compared the attributes of acute phase macrophages (APMΦs) with those of macrophages from uninfected individuals and resolution-phase macrophages (RPMΦs). Using a previously described hemolectin-driven GFP (*HmlGal4 >UAS* eGFP) (*Jung et al., 2005*), we isolated *Drosophila* adult macrophages (approximately 15,000 cells/replicate) and analyzed the metabolic and transcriptional responses that are induced within these cells upon infection (*Figure 1B*). Our approach revealed that *Drosophila* macrophages respond to the acute phase of bacterial infection by increasing glucose uptake, elevating glycolytic flux, and producing lactate. Moreover, as in mammals, the

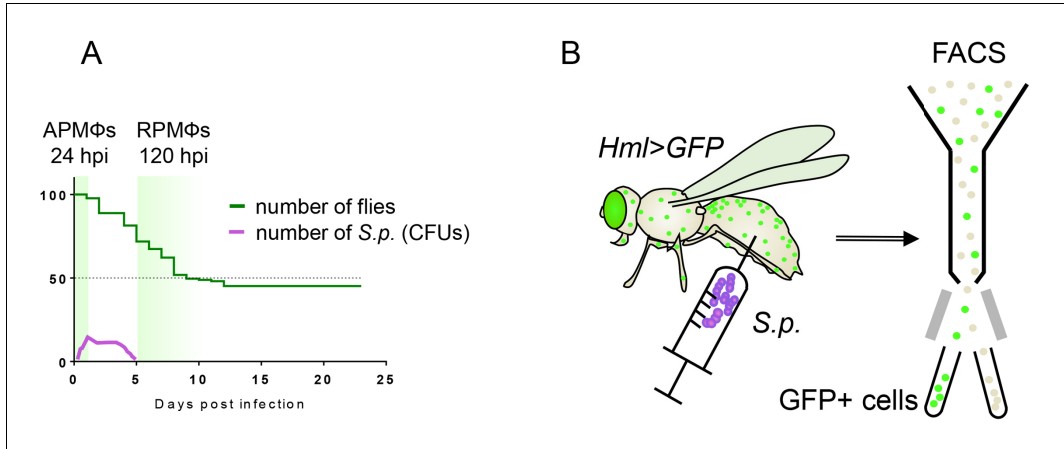

**Figure 1.** Graphical representation of the experimental approach. (**A**) The natural progress of streptococcal infection, with highlighted sampling times during the acute and resolution phases of infection. The Y axis indicates the percentage of surviving adults. (**B**) The approach used to isolate hemocytes, which are subsequently assayed for gene expression and enzymatic activities. Macrophages sorted from flies at the respective time points post-infection represent acute-phase macrophages (APMΦs; 24 hpi) and resolution-phase macrophages (RPMΦs; 120 hpi). Control flies were analyzed at the same time points after receiving injection of phosphate-buffered saline (PBS). hpi, hours post-infection; FACS, fluorescence-activated cell sorting; S.p., *Streptococcus pneumoniae*.
DOI: https://doi.org/10.7554/eLife.50414.003

activation and maintenance of AG within *Drosophila* macrophages depend on Hif1α, and require elevated Ldh activity. We also demonstrate that the induction of AG within *Drosophila* macrophages leads to a change in systemic carbohydrate metabolism. Overall, our findings demonstrate that *Drosophila* macrophages must induce both autonomous and systemic changes in carbohydrate metabolism to mount a proper bactericidal function and to resist infection.

## Results

### *Drosophila* macrophages undergo a metabolic shift to aerobic glycolysis during the acute phase of bacterial infection

Since the bactericidal function of phagocytic cells is connected with AG in mice (*Mills et al., 2000*), we analyzed *Drosophila* macrophages for the occurrence of AG hallmarks, such as increased glucose uptake, an increase in glycolytic flux, and the generation of an NADH pool facilitating the Ldh-mediated reduction of pyruvate to lactate (*Langston et al., 2017*). The distribution of fluorescently labeled deoxyglucose (NBDG) in an organism, frequently used in cancer research, reflects the competitive potential of tissues in glucose internalization (*Cox et al., 2015*). We tested the effect of immune response activation on glucose distribution among tissues in *Drosophila* by feeding the infected or control flies with NBDG during a 24-hr period before the signal detection. Infected flies displayed prominent NBDG accumulation in APMΦs compared to other tissues, which is in contrast to the distribution of NBDG seen in uninfected controls or in flies fed during the resolution phase of infection, which displayed no such accumulation (*Figure 2A,B*). These results indicate an increased potential of phagocytosing macrophages to consume glucose in direct competition with other tissues during the acute phase of bacterial infection.

The increased NBDG uptake by macrophages was further supported by gene expression analysis, which revealed that the transcription of genes encoding both glycolytic enzymes and LDH, but not TCA cycle enzymes, was significantly upregulated in APMΦs (*Figure 2C*). Moreover, these changes in glycolytic genes were restricted to the acute phase of infection as most glycolytic genes returned to a basal level of expression during the resolution phase, whereas hexokinase and enolase (similarly to all analyzed TCA cycle genes) even showed decreased expression (*Figure 2C* and *Figure 2—figure supplement 1*), which can be ascribed to the global suppression of metabolism in these cells.

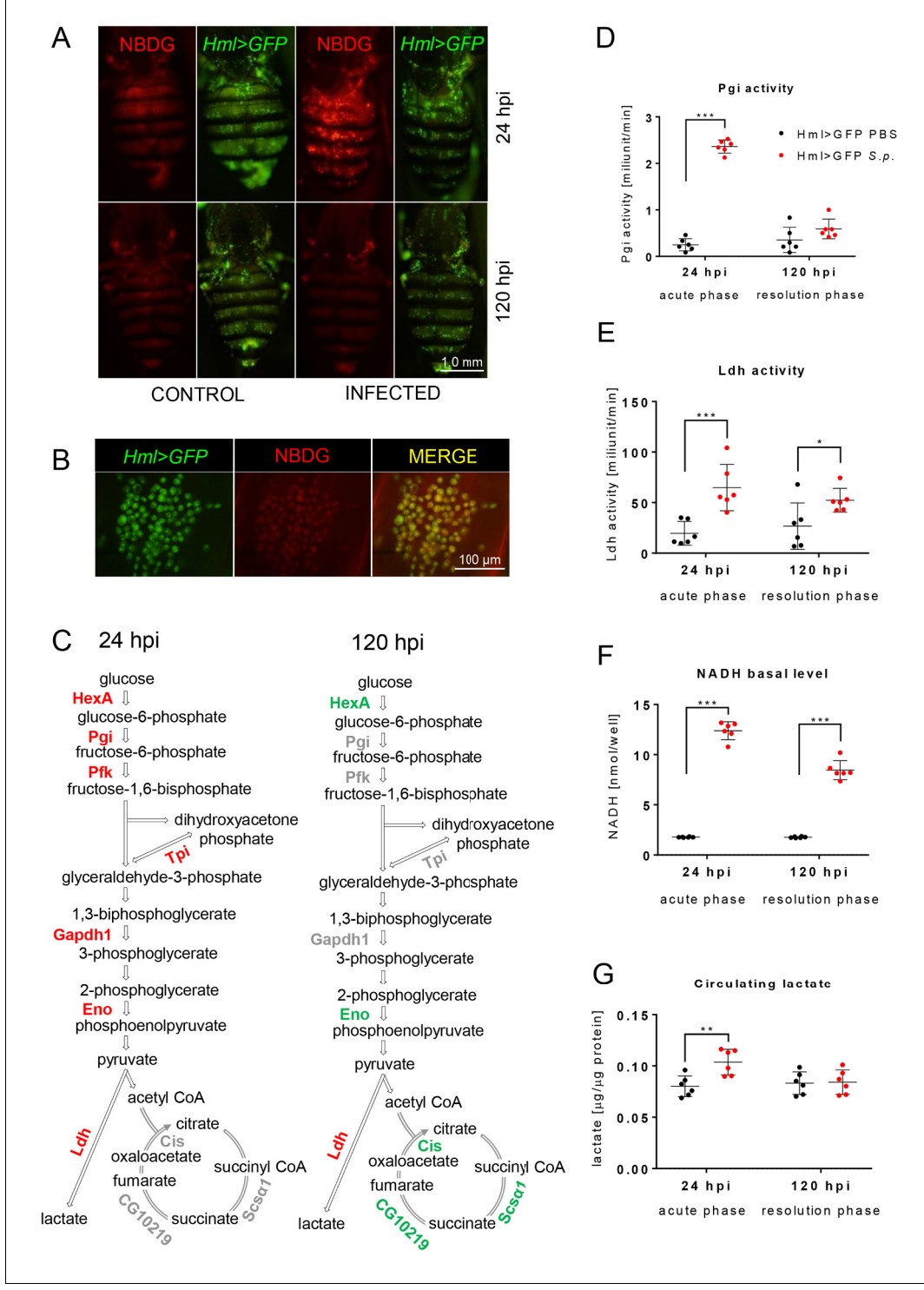

**Figure 2.** Streptococcal infection enhances glycolysis in acute-phase macrophages. (**A–B**) Fluorescent images of the dorsal view of the abdomens of infected and control (both Hml >GFP) flies at 24 and 120 hpi, showing NBDG the distribution among the tissues (**A**) and at a higher magnification (**B**). Images represent a minimum of ten observations of a similar pattern. (**C**) Scheme of glycolysis and the TCA cycle, highlighting significant changes in the quantified expression of the indicated genes at 24 and 120 hpi. The expression levels of the mRNA were measured relative to that of the ribosomal protein 49 (rp49), and the statistical significance (p<0.05) was tested using ANOVA (for data see *Figure 2—figure supplement 1*). Upregulated genes are shown in red, downregulated genes in green; gray indicates no statistically significant difference. (**D–F**) Enzymatic activities of

*Figure 2 continued on next page*

*Figure 2 continued*

phosphoglucose isomerase (Pgi) (D) and lactate dehydrogenase (Ldh) (E), as well as the level of NADH (F), at 24 and 120 hpi measured in the homogenate of hemocytes isolated from infected and control flies. The levels of enzymatic activity and NADH concentration were normalized per ten thousand cells per sample. (G) The concentration of circulating lactate measured in the hemolymph of infected and control flies at 24 and 120 hpi. In all plots (D–G), individual dots represent biological replicates. Values are mean ± SD, asterisks mark statistically significant differences (*p<0.05; **p<0.01; ***p<0.001).
DOI: https://doi.org/10.7554/eLife.50414.004

The following source data and figure supplements are available for figure 2:

**Source data 1.** Metabolic characterization of macrophages post-infection.
DOI: https://doi.org/10.7554/eLife.50414.007
**Figure supplement 1.** Gene expression of glycolytic enzymes is increased in acute-phase macrophages.
DOI: https://doi.org/10.7554/eLife.50414.005
**Figure supplement 1—source data 1.** Expression of metabolic genes in macrophages post-infection.
DOI: https://doi.org/10.7554/eLife.50414.006

Overall, these results indicate that macrophages specifically upregulate glucose metabolism in response to *S. pneumoniae* infection.

Increased glucose uptake and expression of glycolytic genes, including *Ldh*, suggest an increased glycolytic flux and preferential reduction of pyruvate to lactate in APMΦs. To confirm this, we measured the enzymatic activity of LDH, as an enzyme responsible for the diversion of pyruvate from TCA, and phosphoglucose isomerase (Pgi), as a glycolytic enzyme representative. In agreement with the expression data, Pgi enzymatic activity was significantly increased in APMΦs compared to control and compared to the situation observed during the resolution phase of infection (*Figure 2D*). The activity of Ldh increased not only in APMΦs but also in RPMΦs (*Figure 2E*). Moreover, the observed increase in Ldh activity was directly correlated with increased lactate production in vivo, as the hemolymph of infected individuals during both the acute and the resolution phases of infection contained significantly elevated lactate levels as compared to controls (*Figure 2G*). Overall, our results demonstrate that *Drosophila* macrophages respond to *S. pneumoniae* infection by upregulating lactate production.

The primary reason why cells produce lactate as a byproduct of AG is to maintain $NAD^+$/NADH redox balance. High levels of glycolytic flux produce excess NADH as a result of glyceraldehyde-3-phosphate dehydrogenase 1 (Gapdh1) activity (*Olenchock et al., 2017*). Consistently, we observed that NADH levels were significantly increased in APMΦs and, to a lesser extent, in RPMΦs when compared with controls (*Figure 2F*). When considered in the context of gene expression and enzyme activity assays, these results support a model in which activated *Drosophila* macrophages undergo a dramatic metabolic remodeling towards AG during bacterial infection.

## Hif1α and Ldh activities are increased in *Drosophila* macrophages during the acute phase of infection

Since Hif1α can induce AG in both murine and *Drosophila* cells (*Peyssonnaux et al., 2007*; *Herranz and Cohen, 2017*; *Eichenlaub et al., 2018*), we examined the possibility that this transcription factor also promotes glucose catabolism within activated macrophages. Although Hif1α is known to be expressed continuously in almost all tissues and regulated predominantly at the post-translational level, we observed that *Hif1α* mRNA was significantly elevated in APMΦs (*Figure 3D*). To determine whether this increase correlates with the elevated expression of Hif1α target genes, we used a transgenic β-galactosidase reporter under the control of a HRE (*HRE-LacZ*), which is primarily induced by HIF1α (*Lavista-Llanos et al., 2002*) although the involvement of other transcription factors cannot be entirely excluded. Although some cells exhibited *HRE-LacZ* expression in uninfected individuals, the number of β-galactosidase-positive macrophages rose dramatically in flies during the acute phase of infection (*Figure 3A*). These results suggest that Hif1α activity is increased in APMΦs and confirms the previously reported expression pattern of glycolytic genes (see *Figure 2—figure supplement 1A–F*).

As increased lactate production is a hallmark of AG, we examined *Ldh* expression in macrophages using a transgene that expresses a Ldh-mCherry fusion protein from an endogenous *Ldh* promoter.

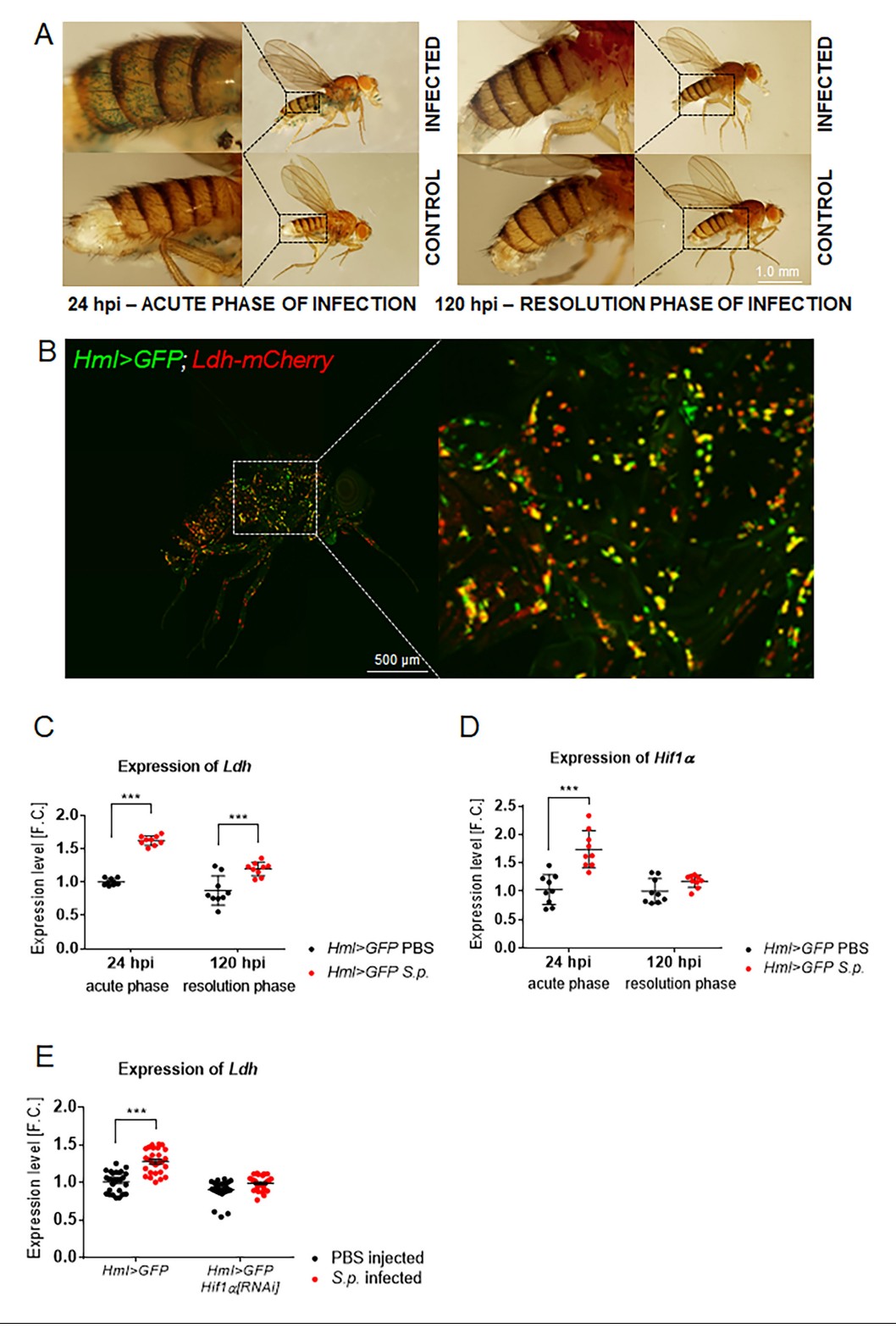

**Figure 3.** Macrophage-specific activities of Hif1α and Ldh increase upon infection. (**A**) X-gal staining of infected and control flies bearing the HRE-LacZ reporter construct. Images represent a minimum of ten observations of a similar pattern. (**B**) An uninfected Hml >GFP, Ldh-mCherry adult fly (24 hpi) shows localization of the Ldh reporter activity (red) in many of the immune cells (green). The image is a Z-stack at maximal projection of 25 confocal slices. (**C, D**) Expression of Ldh (**C**) and Hif1α (**D**) mRNAs in hemocytes isolated from infected and control flies

*Figure 3 continued on next page*

*Figure 3 continued*

(both Hml >GFP; 24 and 120 hpi). (**E**) Expression of Ldh mRNA in hemocytes of infected and control Hml >GFP flies with and without a hemocyte-specific knockdown of Hif1α at 24 hpi. In all plots (**C–E**), expression levels, normalized against rp49, are given as fold change (F.C.) relative to levels in PBS-injected Hml >GFP controls (24 hpi), which were arbitrarily set to 1. Individual dots represent biological replicates. Values are mean ± SD, asterisks mark statistically significant differences (*p<0.05; **p<0.01; ***p<0.001).
DOI: https://doi.org/10.7554/eLife.50414.008
The following source data is available for figure 3:

**Source data 1.** Expression pattern of Hif1α and Ldh genes.
DOI: https://doi.org/10.7554/eLife.50414.009

The expression of Ldh-mCherry in adult flies harboring the *HmlGal4 >UAS* eGFP reporter revealed that macrophages from uninfected adults expressed *Ldh* at levels that markedly exceeded the expression of this reporter in other tissues. This perhaps indicates that these cells are primed to generate lactate prior to infection (*Figure 3B*), as the Ldh-mCherry pattern did not change significantly after infection (data not shown). *Ldh* expression, however, was significantly upregulated in APMΦs (*Figure 3C*), further supporting our observation that *S. pneumoniae* induces Ldh activity (*Figure 2E*), which is in agreement with elevated NADH levels (*Figure 2G*). The regulation of *Ldh* expression by Hif1α in activated immune cells was verified by knocking down *Hif1α* expression in macrophages 24 hr before infection (*Hml >Hif1α[RNAi]*). This strategy not only reduced *Hif1α* expression within APMΦs (*Figure 4—figure supplement 1G*), but also led to the loss of the ability to increase *Ldh* expression in APMΦs, indicating that Hif1α is essential for the elevated Ldh activity in APMΦs (*Figure 3E*).

## Hif1α promotes aerobic glycolysis in *Drosophila* macrophages during bacterial infection

To determine whether the observed increase in Hif1α activity is necessary to trigger AG in stimulated macrophages, we used *Hml >Hif1α[RNAi]* and examined the metabolic consequences. This treatment led to the abrogation of the metabolic changes associated with AG. Following infection, APMΦs expressing *Hif1α[RNAi]* did not accumulate NBDG (*Figure 4A*), and failed to show increased expression of glycolytic genes (with the exception of Gpdh1) (*Figure 4B*). Moreover, these *Hml >Hif1α[RNAi]*-expressing cells exhibited no increase in either Pgi or Ldh enzyme activity and displayed decreased NADH levels when compared with controls (*Figure 4D,E,F*). These results indicate that Hif1α activity is essential for inducing AG in macrophages during the immune response.

As a complement to these cell-specific studies of *Hif1α*, we also used *Hml-Gal4* driving *UAS-Ldh [RNAi] (Hml > Ldh[RNAi])* to reduce *Ldh* expression within APMΦs. Intriguingly, although this approach successfully reduced Ldh activity in macrophages (*Figure 4G*), the metabolic consequences were relatively mild. Within APMΦs, *Hml > Ldh[RNAi]* did not disrupt NBDG uptake and Pgi activity remained elevated (*Figure 4C and I*). Twenty-four hours after infection, however, we observed that NADH in *Hml > Ldh[RNAi]* macrophages failed to increase to the levels observed in infected controls (*Figure 4H*), thus revealing that increased Ldh activity is required for full metabolic reprograming of *Drosophila* macrophages in response to bacterial infection.

## Hif1α-mediated aerobic glycolysis in APMφs causes systemic metabolic changes

As we have shown previously (*Bajgar and Dolezal, 2018*), the systemic metabolic adaptation of carbohydrate metabolism is intimately linked to the effective function of the immune system during streptococcal infection. Therefore, we focused on the characterization of systemic carbohydrate metabolism during the acute phase of infection in *Hml >Hif1α[RNAi]* and *Hml >Ldh[RNAi]* flies (*Figure 5*). Both control genotypes underwent the expected metabolic response during the acute phase of streptococcal infection: a significantly raised level of circulating glucose was accompanied by a strong depletion of glycogen stores in tissues. The *Hif1α* silencing completely suppressed the infection-induced changes in carbohydrate metabolism, but infected *Hml >Ldh[RNAi]* flies still significantly increased circulating glucose, albeit to a lesser extent than in the infected controls

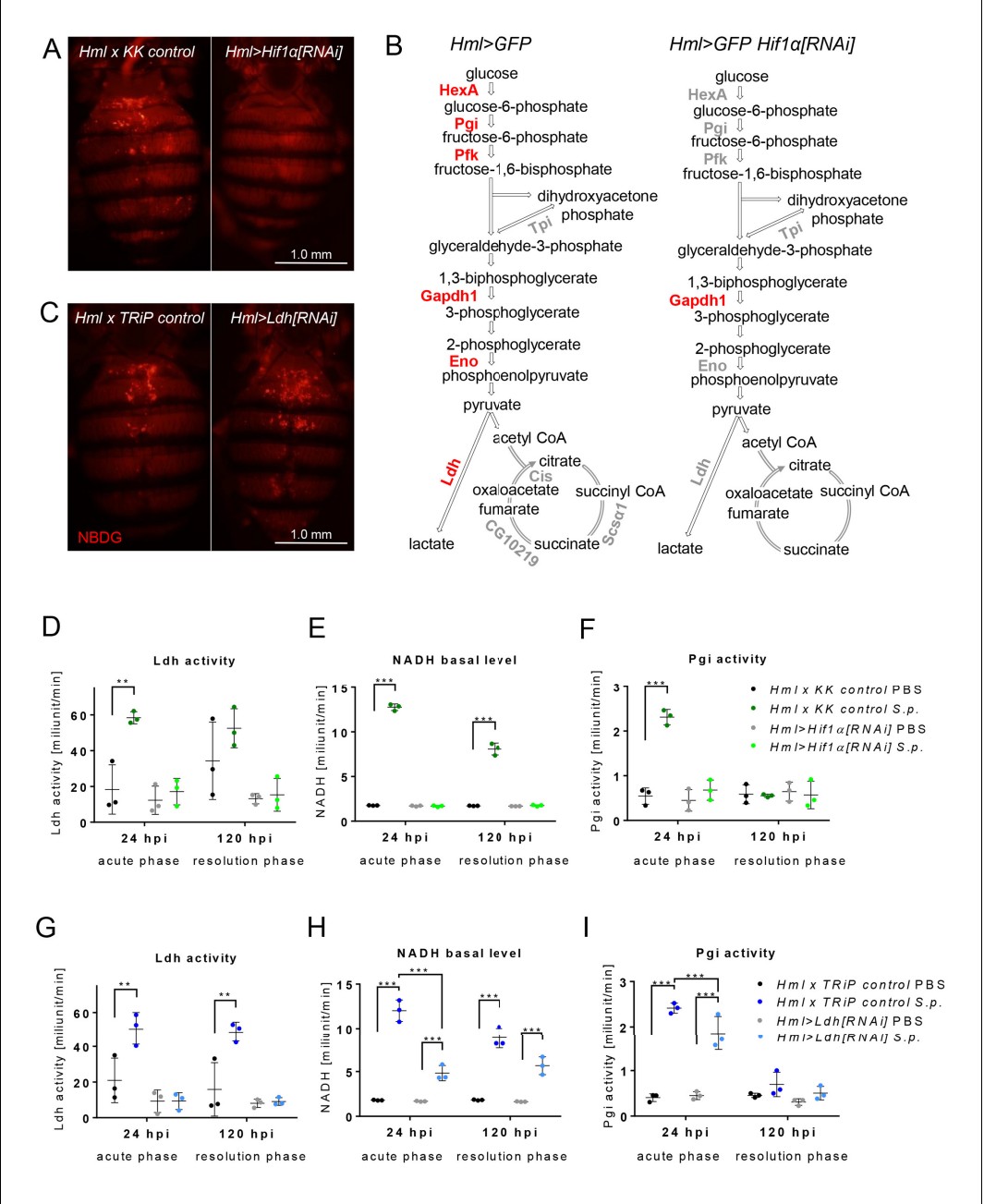

**Figure 4.** Effects of Hif1α and Ldh hemocyte-specific knockdown on macrophage metabolism. (**A**) Dorsal view of the abdomens of S.p.-infected flies (24 hpi) showing the distribution of the fluorescent NBDG probe. Controls (left) are compared to flies subjected to hemocyte-specific knockdown of Hif1α. Images represent a minimum of ten observations of a similar pattern. (**B**) Schematic representation of the expression of genes encoding metabolic enzymes in the hemocytes of infected control flies (left) and of flies with Hif1α hemocyte-specific knockdown (right) at 24 hpi. The expression levels of the mRNAs were measured relative to that of rp49, and the statistical significance (p<0.05) was tested using ANOVA (for data see *Figure 4—figure supplement 1*). Upregulated genes are shown in red; gray indicates no statistically significant difference. (**C**) Dorsal view of the abdomens of S.p.-infected flies (24 hpi) showing the distribution of the fluorescent NBDG probe. Controls (left) are compared to flies subjected to hemocyte-specific knockdown of Ldh. Images represent a minimum of ten observations of a similar pattern. (**D–F**) Enzymatic activity of Ldh (**D**), level of NADH (**E**), and enzymatic activity of Pgi (**F**) at 24 and 120 hpi measured in lysates of hemocytes isolated from infected and non-infected control flies and from flies with Hif1α hemocyte-specific knockdown. (**G–I**) Enzymatic activity of Ldh (**G**), level of NADH (**H**), and enzymatic activity of Pgi (**I**) at 24 and 120 hpi measured in lysates of hemocytes isolated from infected and non-infected control flies and

*Figure 4 continued on next page*

*Figure 4 continued*
from flies with Ldh hemocyte-specific knockdown. In all plots (**D–I**), the enzyme activities and NADH concentrations were normalized per ten thousand cells per sample. Individual dots represent biological replicates. Values are mean ± SD, asterisks mark statistically significant differences (*p<0.05; **p<0.01; ***p<0.001).
DOI: https://doi.org/10.7554/eLife.50414.010
The following source data and figure supplements are available for figure 4:

**Source data 1.** Effect of macrophage-specific Hif1α knockdown on metabolic features of macrophages.
DOI: https://doi.org/10.7554/eLife.50414.013
**Figure supplement 1.** Expression of genes encoding glycolytic enzymes is not increased in acute-phase macrophages with Hif1α knock-down.
DOI: https://doi.org/10.7554/eLife.50414.011
**Figure supplement 1—source data 1.** Effect of macrophage-specific Hif1α knockdown on expression of metabolic genes.
DOI: https://doi.org/10.7554/eLife.50414.012

(*Figure 5A*). Although the glycogen stores appeared to be lowered in *Hml >Ldh[RNAi]* flies upon infection, the decrease was statistically insignificant (*Figure 5B*). Importantly, the macrophage-specific knockdown of either *Hif1α* or *Ldh* suppressed the occurrence of an infection-induced increase in circulating lactate titer (*Figure 5C*). These results show that APMΦs are prominent lactate producers during the acute phase of the infection, and suggest that only full activation of APMΦs with Hif1α-induced metabolic changes leads to reprograming of systemic carbohydrate metabolism.

## Hif1α- and Ldh-mediated metabolic remodeling of APMΦs is essential for mounting a successful immune response

Our results suggest that *Drosophila* macrophages activate AG and systemic metabolic changes in order to mount a successful immune response. In support of this hypothesis, we observed a significant decrease in the viability of adult flies expressing either *Hml >Hif1α[RNAi]* or *Hml >Ldh[RNAi]* following *S. pneumoniae* infection. By 72 hr post infection, 25% of *Hml >Hif1α[RNAi]* flies died compared to 7% of controls, and the medium time to death (MTD) in *Hml >Hif1α[RNAi]* flies was 10 days compared to 23 days in controls (*Figure 6A*). Moreover, pathogen load in *Hml >Hif1α[RNAi]* flies was substantially elevated when compared with that in controls at the second and third day post-infection (*Figure 6C*). We observed similar effects in *Hml >Ldh[RNAi]* flies, in which *S. pneumoniae* infection resulted in a decreased survival rate, a MTD of 9 days relative to the 18 days observed in controls, and elevated bacterial load during days 2 and 3 post-infection (*Figure 6B and D*). These results reveal that Hif1α and Ldh serve essential roles in both survival of infection and bacterial killing, and demonstrate how shift towards AG associated with systemic metabolic changes in activated macrophages is required to mount a successful immune response.

## Discussion

Mammalian macrophages stimulated by bacteria have been shown to rewire their metabolism temporarily towards AG in order to develop an adequate bactericidal response (*Olenchock et al., 2017*; *Nonnenmacher and Hiller, 2018*; *Browne et al., 2013*). Although well established in mammals, such metabolic adaptation has not been experimentally tested in insect macrophages to date. We show here that *Drosophila* macrophages that are activated by bacterial infection undergo a dramatic remodeling of cellular metabolism. We demonstrate that acute-phase macrophages exhibit hallmarks of AG, such as elevated uptake of glucose, increased expression and activity of glycolytic genes, elevation of NADH levels, and preferential LDH-mediated conversion of pyruvate to lactate. Through macrophage-specific gene knockdown, we identified Hif1α to be essential for the induction of increased glycolytic flux as well as for the increased activity of LDH. Both Hif1α and Ldh are necessary for the full development of infection-induced changes in systemic carbohydrate metabolism and for resistance to bacterial infection.

A major takeaway of our work is that the cellular response to bacterial infection is an energetically challenging process that imposes significant metabolic demands upon the host. Our findings demonstrate that *Drosophila* macrophages meet these metabolic demands by inducing AG during the

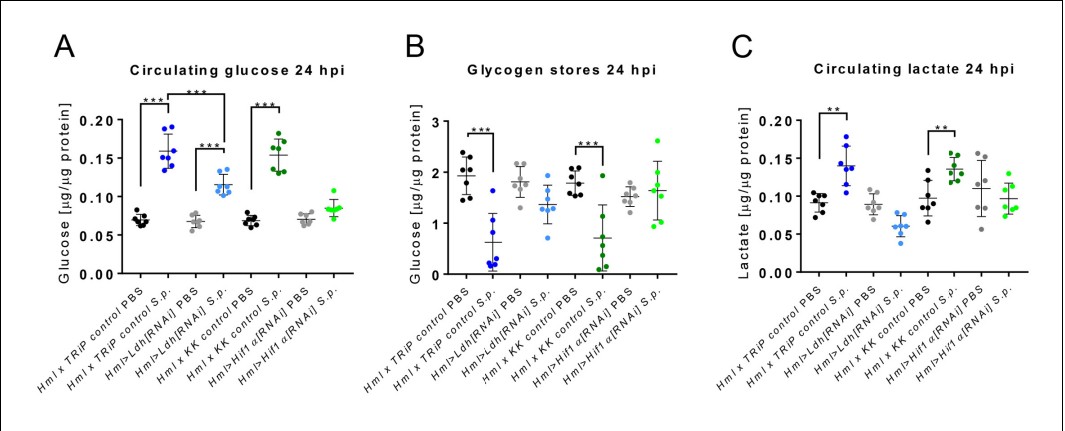

**Figure 5.** Systemic effects of Hif1α and Ldh hemocyte-specific knockdown. (**A–C**) The concentration of circulating glucose (**A**), glycogen stores (**B**) and circulating lactate (**C**) in infected and non-infected flies with Hif1α or Ldh hemocyte-specific knockdown and their respective controls at 24 hpi. The concentrations of metabolites were normalized to the amount of proteins in each sample. Individual dots in the plot represent biological replicates. Values are mean ± SD, asterisks mark statistically significant differences (*p<0.05; **p<0.01; ***p<0.001).
DOI: https://doi.org/10.7554/eLife.50414.014
The following source data is available for figure 5:

**Source data 1.** Effect of macrophage-specific Hif1α and Ldh knockdown on systemic carbohydrate metabolism.
DOI: https://doi.org/10.7554/eLife.50414.015

acute phase of *S. pneumoniae* infection, as evidenced by the increased expression of glycolytic enzyme genes and elevated NADH levels. This increase in LDH enzyme activity in the absence of elevated TCA cycle activity suggests that macrophages preferentially convert pyruvate to lactate and is consistent with the elevated concentration of lactate observed in hemolymph. However, we find that this metabolic adaptation is temporary, as AG is terminated during the resolution phase of infection. This latter observation is important because it reveals that macrophages temporally regulate metabolic flux throughout an infection and because it establishes *Drosophila* as a powerful model for exploring the molecular mechanisms that control immune cell metabolism.

Our findings also extend the similarities between fly and mammalian models of macrophage polarization, as we identified Hif1α and Ldh as being crucial for the establishment and maintenance of AG in acute-phase macrophages. The importance of these factors is demonstrated by the macrophage-specific *Hif1α* knockdown experiment described above, in which many of the hallmark characteristics of AG, including expression of the *Ldh* gene, were abolished. This finding highlights the conserved and ancient role for Hif1α in regulating the switch between glycolytic and oxidative metabolism (*Webster, 2003*), and suggests that this function evolved as a means of allowing cells to adapt quickly to changing physiological conditions and cell-specific metabolic needs. The role of Hif1α in regulating this switch is of significant interest because, although this transcription factor is classically associated with the response to hypoxia, our study adds to the growing list of examples in which Hif1α remodels cellular metabolism in the context of cell proliferation, activation, and competition, even under normoxic conditions (*Miyazawa and Aulehla, 2018*). Moreover, our finding is particularly intriguing in light of the fact that Hif1α also serves a key role in promoting AG in neoplastic tumor cells (*Herranz and Cohen, 2017*; *Eichenlaub et al., 2018*; *Wang et al., 2016*). Therefore, our studies of fly macrophages provide a new in vivo system in which we can study how Hif1α promotes cell activity by modulating central carbon metabolism.

While Hif1α drives AG in *Drosophila* macrophages via transcriptional regulation of target genes, the role of Ldh in these cells is more complicated. Although acute-phase macrophages still consume more glucose upon *Ldh* knockdown, these cells exhibit significantly lower Pgi activity and NADH levels, and the titer of circulating lactate also drops. Our results suggest that even though Ldh acts only at the last step of AG, its role is essential for full metabolic reprograming and efficient function of immune cells. *Drosophila* Ldh, like its mammalian ortholog, is responsible for the reduction of

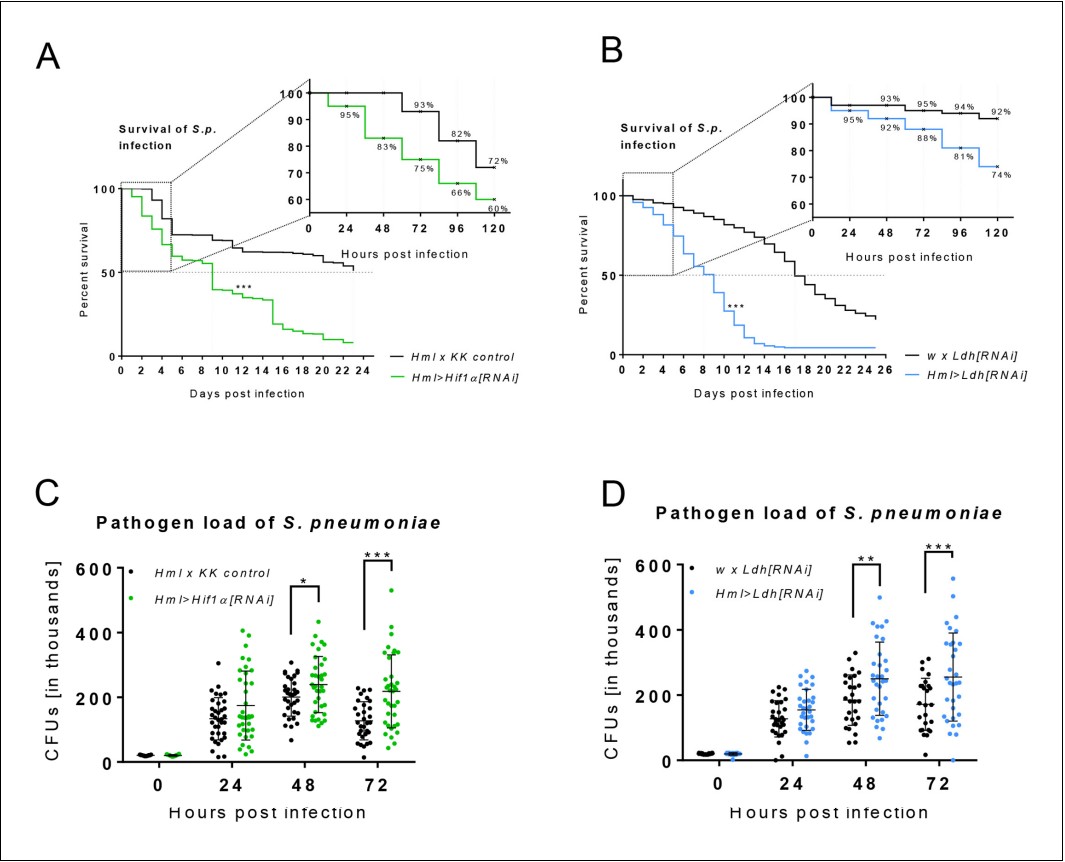

**Figure 6.** Effects of Hif1α and Ldh hemocyte-specific knockdown on resistance to infection. (A–B) The survival rate of infected flies of the control genotype and of flies with hemocyte-specific Hif1α (A) and Ldh (B) knockdown. Vertical dotted lines denote medium time to death for each genotype; survival rate during the first 120 hr is shown in detail. Three independent experiments were performed and combined into one survival curve. The average number of individuals per replicate was more than 500 for each genotype. (C, D) Colony forming units (CFUs) obtained from infected flies of control genotype and from flies with hemocyte-specific Hif1α (C) and Ldh (D) knockdown at 0, 24, 48, and 72 hpi. Individual dots in the plot represent the number of bacteria raised from one individual. The data show results merged from three independent biological replicates.

DOI: https://doi.org/10.7554/eLife.50414.016

The following source data is available for figure 6:

**Source data 1.** Effect of macrophage-specific Hif1α and Ldh knockdown on the resistance to bacterial infection.
DOI: https://doi.org/10.7554/eLife.50414.017

pyruvate to lactate, which is linked with the regeneration of NAD$^+$ from NADH. However, this single reaction has an immense impact on cellular metabolism. Both the accumulation of pyruvate and the lack of NAD$^+$ can become limiting in cells with high glycolytic flux (*Olenchock et al., 2017*). In addition, Ldh-dependent removal of cytosolic pyruvate was recently found to be essential to prevent pyruvate entry into mitochondria and a subsequent change of TCA cycle course (*Eichenlaub et al., 2018*; *Wang et al., 2016*).

Although not addressed in our study, changes in mitochondrial metabolism are also closely associated with AG and should be the focus of future studies of activated *Drosophila* macrophages. The interconnection between the transcriptional activity of Hif1α and the change of mitochondrial metabolism in *Drosophila* has recently been elucidated. The activation of several direct targets of Hif1α transcriptional activity leads to an inhibition of the classical course of the TCA cycle (*Wang et al., 2016*; *Eichenlaub et al., 2018*). One of the well-understood mechanisms is the prevention of pyruvate entry into the TCA cycle, which is caused by increased kinase activity of pyruvate dehydrogenase kinase 1 (PDK1). PDK1-mediated phosphorylation of pyruvate

dehydrogenase (PDH) directly inhibits its enzymatic function, which is essential for pyruvate conversion to acetyl-CoA (*Wang et al., 2016*). This event causes a cytoplasmic accumulation of TCA cycle intermediates and thus promotes a secondary wave of Hif1α stabilization through inhibition of prolyl hydroxylase dehydrogenase (PHD) under normoxic conditions (*Koivunen et al., 2007*; *Freije et al., 2012*). The change in the TCA cycle is further needed for mitochondrial production of ROS that are transferred to the phagolysosome for bacterial killing (*Forrester et al., 2018*; *Williams and O'Neill, 2018*).

We further demonstrate that both Hif1α and Ldh are crucial not only for full macrophage activation, but also for the bactericidal function of the immune cells, with the rearrangement of macrophage metabolism towards AG being essential for resistance to infection and host survival. An important aspect of AG is the functional dependence of macrophages on sufficient supply of external energy resources, as demonstrated in both mammalian and insect phagocytes (*Bajgar and Dolezal, 2018*; *Anderson et al., 1973a*; *Anderson et al., 1973b*; *Langston et al., 2017*; *Newsholme et al., 1986*) and documented here by increased consumption of glucose. Immune cells therefore generate systemic factors to secure sufficient supply of nutrients by altering the function of other organs and by regulating systemic metabolism (*Bajgar et al., 2015*; *Bajgar and Dolezal, 2018*). While the identification of specific signaling factors is beyond the scope of this work, there are several candidate molecules in *Drosophila* that are known to be produced by activated macrophages as a reaction to the metabolic state of the cell. Although it is likely that multiple factors will be involved in this process, we can presume that these factors will reflect the metabolic state of the cells (e.g., extracellular adenosine), or that they will be linked to the transcriptional program that causes the switch towards AG (e.g., Imaginal morphogenesis protein late 2 (ImpL2)). In our previous work, we showed that the systemic metabolic switch upon infection depends on extracellular adenosine, which is produced by the activated immune cells (*Bajgar and Dolezal, 2018*; *Bajgar et al., 2015*). The production of adenosine directly reflects a metabolic state of the cell, such as increased consumption of ATP (*Worku and Newby, 1983*), as well as the accelerated occurrence of methylation events (*German et al., 1983*; *Wu et al., 2005*). Expression of ImpL2 was shown to be regulated by Hif1α (*Li et al., 2013*), and as ImpL2 was previously identified as a mediator of cancer-induced loss of energy reserves in flies due to its anti-insulin role (*Kwon et al., 2015*; *Figueroa-Clarevega and Bilder, 2015*), it could represent another link between AG in macrophages and changes in systemic metabolism that ensure sufficient supply of energy resources.

Finally, our findings raise an interesting question regarding the links between AG and the ability of immune cells to respond quickly to infection. Recent studies of mammalian macrophage metabolism revealed that AG is essential for the development of innate immune memory - called trained immunity (*Netea et al., 2016*). The mechanism of trained immunity relies on chromatin remodeling by epigenetic factors that enable cells to react with higher efficiency in response to re-infection by a particular pathogen (*Kim et al., 2019*). As many chromatin remodeling enzymes need cofactors (such as acetyl-CoA, NAD⁺, α-KG) for the remodeling of the epigenetic landscape, their function can be influenced by the metabolic state of the cell. Induction of AG leads to the accumulation of many cofactors that are essential for a proper function of these enzymes (*Kim et al., 2019*; *Baardman et al., 2015*). The concept of trained immunity not only is valid for mammals, but is rather present in many invertebrate clades (where it is called immune priming; *Milutinović and Kurtz, 2016*; *Pham et al., 2007*). Our observation of AG as a characteristic feature of activated *Drosophila* macrophages thus raises a question of its importance for the development of trained immunity in insects and other invertebrates. Taken together, our findings demonstrate how the molecular mechanisms that control AG induction in *Drosophila* macrophages exhibit a surprisingly high level of evolutionary conservation between mammals and insects, thus emphasizing that this metabolic switch is essential for survival of infection and hinting at the potential role for AG in the development of immune memory.

In conclusion, we have shown that infection-induced systemic changes in carbohydrate metabolism are associated with changes of macrophage cellular metabolism, and that both can be affected by macrophage-specific *Hif1α* and *Ldh* knockdown. Our data thus link the metabolic state of macrophages with the systemic metabolic changes. On the basis of our previous research on the selfish nature of the immune system under challenge (*Straub, 2014*), we envision that the shift in the cellular metabolism of macrophages leads to the production of signals that alter the systemic metabolism, thereby securing the sufficient energy supply necessary to allow the macrophages to fight the

infection. By linking the induction of macrophage polarization with systemic metabolism and systemic outcomes in vivo, our experimental system can aid future research towards better understanding of the immune system and of diseases related to its malfunction.

# Materials and methods

## Key resources table

| Reagent type (species) or resource | Designation | Source reference | Identifier | Additional information |
|---|---|---|---|---|
| Strain, strain background (*Streptococcus pneumoniae*) | EJ1 strain | Provided by David Schneider | | Dilution 20,000 units |
| Chemical compound, drug | TRIzol Reagent | Invitrogen | Cat# 15-596-018 | |
| Chemical compound, drug | Superscript III Reverse Transcriptase | Invitrogen | Cat# 18080044 | |
| Chemical compound, drug | 2x SYBR Master Mix | Top-Bio | Cat# T607 | |
| Chemical compound, drug | 2-NBDG | Thermo Fisher Scientific | Cat# N13195 | |
| Chemical compound, drug | X-gal | Sigma | Cat# B4252 | |
| Commercial assay, kit | Glucose (GO) Assay Kit | Sigma | Cat# GAGO20-1KT | |
| Commercial assay, kit | Bicinchoninic Acid Assay Kit | Sigma | Cat# BCA1 | |
| Commercial assay, kit | Lactate Assay Kit | Sigma | Cat# MAK064 | |
| Commercial assay, kit | Lactate Dehydrogenase Activity Assay Kit | Sigma | Cat# MAK066 | |
| Commercial assay, kit | Phosphoglucose Isomerase Colorimetric Assay Kit | Sigma | Cat# MAK103 | |
| Genetic reagent (*Drosophila melanogaster*) | HmlG4G80: w*; HmlΔ-Gal4*; P{tubPGal80ts}* | Cross made in our laboratory by Tomas Dolezal | | |
| Genetic reagent (*D. melanogaster*) | Hml > GFP: w; HmlΔ-Gal4 UAS-eGFP | Provided by Bruno Lemaitre | | |
| Genetic reagent (*D. melanogaster*) | Hif1α[RNAi]: P{KK110834}VIE-260B | Vienna Drosophila Resource Center | VDRC: v106504 | FBst0478328 |
| Genetic reagent (*D. melanogaster*) | TRiP control: y(1) v(1); P{y[+t7.7]=CaryP}attP2 | Bloomington Drosophila Stock Center | BDSC: 36303 | FBst0036303 |
| Genetic reagent (*D. melanogaster*) | KK control: y, w[1118];P{attP,y[+],w[3']} | Bloomington Drosophila Stock Center | BDSC: 60100 | FBst0060100 |
| Genetic reagent (*D. melanogaster*) | Ldh[RNAi]: y(1) v(1); P{y[+t7.7] v[+t1.8]=TRiP. HMS00039}attP2 | Bloomington Drosophila Stock Center | BDSC: 33640 | FBst0033640 |
| Genetic reagent (*D. melanogaster*) | HRE-LacZ: HRE-HRE-CRE-LacZ | Provided by Pablo Wappner (*Lavista-Llanos et al., 2002*) | | |
| Genetic reagent (*D. melanogaster*) | Ldh-mCherry | Provided by Jason Tennessen | | |

*Continued on next page*

*Continued*

| Reagent type (species) or resource | Designation | Source reference | Identifier | Additional information |
|---|---|---|---|---|
| Genetic reagent (D. melanogaster) | w: w[1118] | Genetic background based on CantonS | | |
| Sequence-based reagent | Cis forward: 5'TTCGATTGACTCCAGCCTGG3' | KRD | CG14740 | FBgn0037988 |
| Sequence-based reagent | Cis reverse: 5'AGCCGGGAACCACCTGTCC3' | KRD | CG14740 | FBgn0037988 |
| Sequence-based reagent | Ldh forward: 5'CAGAGAAGTGGAACGAGCTG3' | KRD | CG10160 | FBgn0001258 |
| Sequence-based reagent | Ldh reverse: 5'CATGTTCGCCCAAAACGGAG3' | KRD | CG10160 | FBgn0001258 |
| Sequence-based reagent | Eno forward: 5'CAACATCCAGTCCAACAAGG3' | KRD | CG17654 | FBgn0000579 |
| Sequence-based reagent | Eno reverse: 5'GTTCTTGAAGTCCAGATCGT3' | KRD | CG17654 | FBgn0000579 |
| Sequence-based reagent | Gapdh1 forward: 5'TTG TGG ATC TTA CCG TCC GC3' | KRD | CG12055 | FBgn0001091 |
| Sequence-based reagent | Gapdh1 reverse: 5'CTCGAACACAGACGAATGGG3' | KRD | CG12055 | FBgn0001091 |
| Sequence-based reagent | HexA forward: 5'ATATCGGGCATGTATATGGG3' | KRD | CG3001 | FBgn0001186 |
| Sequence-based reagent | HexA reverse: 5'CAATTTCGCTCACATACTTGG3' | KRD | CG3001 | FBgn0001186 |
| Sequence-based reagent | Pfk forward: 5'AGCTCACATTTCCAAACATCG3' | KRD | CG4001 | FBgn0003071 |
| Sequence-based reagent | Pfk reverse: 5'TTTGATCACCAGAATCACTGC3' | KRD | CG4001 | FBgn0003071 |
| Sequence-based reagent | Pgi forward: 5'ACTGTCAATCTGTCTGTCCA3' | KRD | CG8251 | FBgn0003074 |
| Sequence-based reagent | Pgi reverse: 5'GATAACAGGAGCATTCTTCTCG3' | KRD | CG8251 | FBgn0003074 |
| Sequence-based reagent | Rp49 forward: 5'AAGCTGTCGCACAAATGGCG3' | KRD | CG7939 | FBgn0002626 |
| Sequence-based reagent | Rp49 reverse: 5'GCACGTTGTGCACCAGGAAC3' | KRD | CG7939 | FBgn0002626 |
| Sequence-based reagent | Hif1α forward: 5'CCAAAGGAGAAAAGAAGGAAC3' | KRD | CG45051 | FBgn0266411 |
| Sequence-based reagent | Hif1α reverse: 5'GAATCTTGAGGAAAGCGATG3' | KRD | CG45051 | FBgn0266411 |
| Sequence-based reagent | CG10219 forward: 5'GAGATCTCCGTGAGTGCGC3' | KRD | CG10219 | FBgn0039112 |
| Sequence-based reagent | CG10219 reverse: 5'CTCCACGCCCCAATGGG3' | KRD | CG10219 | FBgn0039112 |
| Sequence-based reagent | Scsα1 forward: 5'TCACAAGCGCGGCAAGATC3' | KRD | CG1065 | FBgn0004888 |
| Sequence-based reagent | Scsα1 reverse: 5'TTGATGCCCGAATTGTACTCG3' | KRD | CG1065 | FBgn0004888 |
| Sequence-based reagent | Tpi forward: 5'AGATCAAGGACTGGAAGAACG3' | KRD | CG2171 | FBgn0086355 |
| Sequence-based reagent | Tpi reverse: 5'ACCTCCTTGGAGATGTTGTC3' | KRD | CG2171 | FBgn0086355 |
| Software, algorithm | Graphpad Prism | https://www.graphpad.com/ | Graphpad Prism | RRID: SCR_002798 |

*Continued on next page*

*Continued*

| Reagent type (species) or resource | Designation | Source reference | Identifier | Additional information |
|---|---|---|---|---|
| Software, algorithm | Microsoft Excel | https://www.microsoft.com/ | Microsoft Excel | |
| Software, algorithm | Fiji | ImageJ - https://fiji.sc | ImageJ | RRID: SCR_002285 |
| Other | S3e Cell Sorter | BioRad - http://www.bio-rad.com/ | BioRad | |
| Other | Olympus FluoView 1000 | Olympus - https://www.olympus-global.com/ | Olympus | RRID: SCR_017015 RRID: SCR_014215 |
| Other | Olympus SZX12 | Olympus - https://www.olympus-global.com/ | Olympus | |
| Other | Olympus IX71 | Olympus - https://www.olympus-global.com/ | Olympus | |

### *Drosophila melanogaster* strains

Flies were raised on a diet containing cornmeal (80 g/l), agar (10 g/l), yeast (40 g/l), saccharose (50 g/l) and 10% methylparaben (16.7 mL/l) and were kept in a controlled humidity environment with natural 12 hr/12 hr light/dark periods at 25°C, except for those used in temperature-controlled Gal80 experiments. Flies bearing Gal80 were transferred at 29°C 24 hr prior to infection in order to degrade temperature-sensitive Gal 80 protein. Prior to experiments, flies were kept in plastic vials on 0% glucose diet (cornmeal 53.5 g/l, agar 6.2 g/l, yeast 28.2 g/l and 10% methylparaben 16.7 mL/l) for 7 days and transferred into fresh vials every second day without $CO_2$ in order to ensure good condition of the food. Infected flies were kept on 0% glucose diet in incubators at 29°C due to the temperature sensitivity of *S. pneumoniae*. Drosophila Stock Centre in Bloomington provided TRiP control and Ldh[RNAi] flies. Hif1α[RNAi] and KK control flies were obtained from Vienna Drosophila Resource Center. Ldh-mCherry strain was kindly provided by Jason Tennessen, HRE-LacZ by Pablo Wappner and Hml > GFP by Bruno Lemaitre. The w1118 strain has a genetic background based on CantonS.

### Bacterial strain and fly injection

The *S. pneumoniae* strain EJ1 was stored at −80°C in Tryptic Soy Broth (TSB) media containing 16% glycerol. For the experiments, bacteria were streaked onto agar plates containing 3% TSB and 100 µg/mL streptomycin and subsequently incubated at 37°C + 5% $CO_2$ overnight. Single colonies were inoculated into 3 mL of TSB liquid media with 100 µg/mL of streptomycin and 100,000 units of catalase and incubated at 37°C + 5% $CO_2$ overnight. Bacterial density was measured after an additional 4 hr so that it reached an approximate 0.4 OD600. Final bacterial cultures were centrifuged and dissolved in phosphate-buffered saline (PBS) so the final OD reached A = 2.4. The *S. pneumoniae* culture was kept on ice prior to injection and during the injection itself. Seven-day-old males (survival experiments, qPCR assays, measurement of metabolites and enzymatic activity) or females (X-gal staining, NBDG assay) were anaesthetized with $CO_2$ and injected with 50 nL culture containing 20,000 *S. pneumoniae* bacteria or 50 nL of mock buffer (PBS) into the ventrolateral side of the abdomen using an Eppendorf Femtojet Microinjector.

### Pathogen load measurement

Sixteen randomly chosen flies per genotype and treatment were anaesthetized with $CO_2$ and individually homogenized in 200 µL PBS using a motorized plastic pestle. Serial dilutions were plated onto TSB agar plates and incubated at 37°C overnight. The number of colonies was counted at 0, 24, 48 and 72 hpi. Collected data were compared using Tukey's multiple comparisons test in Graphpad Prism software. Sidak's multiple comparison correction was performed.

## Survival analysis

Injected flies were kept at 29°C in vials with approximately 30 individuals per vial and were transferred onto a fresh food every other day. Dead flies were counted daily. At least three independent experiments were performed and combined into one survival curve created in Graphpad Prism software; the individual experiments showed comparable results. Average number of individuals was more than 500 for each genotype. Data were analyzed by Log-rank and Grehan-Breslow-Wilcoxon tests (which gave more weight to deaths at early time points).

## Isolation of hemocytes

GFP-labeled hemocytes were isolated from HmlΔ-Gal4 UAS-eGFP male flies using fluorescence-activated cell sorting (FACS). Approximately 200 flies were anaesthetized with $CO_2$, washed in PBS and homogenized in 600 µL of PBS using a pestle. Homogenate was sieved through a nylon cell strainer (∅ 40 µm). This strainer was then additionally washed with 200 µL of PBS, which was added to the homogenate subsequently. Samples were centrifuged (3 min, 6°C, 3500 RPM) and the supernatant was washed in ice cold PBS after each centrifugation (3x). Prior to sorting, samples were transferred to polystyrene FACS tubes using a disposable bacterial filter (∅ 50 µm, Sysmex) and sorted into 100 µL of TRIzol Reagent (Invitrogen) using a S3TM Cell Sorter (BioRad). Sorted cells were verified by fluorescent microscopy and by differential interference contrast (DIC).

## Gene expression

Sorted hemocytes were homogenized using a DEPC-treated pestle and RNA was extracted by TRIzol Reagent (Invitrogen) according to the manufacturer's protocol. Superscript III Reverse Transcriptase (Invitrogen) and oligo(dT)20 primer was used for reverse transcription. Amounts of mRNA of particular genes were quantified on a CFX 1000 Touch Real-Time Cycler (Bio-Rad) using the TP 2x SYBR Master Mix (Top-Bio) in three technical replicates with the following conditions: initial denaturation for 3 min at 95°C, then amplification for 15 s at 94°C, 30 s at 54°C, 40 s at 72°C for 40 cycles and melting curve analysis at 65–85°C/step 0.5°C. Primer sequences are listed in the Key Resources Table. qPCR data were analyzed with double delta Ct analysis, and expressions or particular genes were normalized to the expression of Ribosomal protein 49 (Rp49) in the same sample. Relative values (fold change) to control were compared and are shown in the graphs. Samples for gene expression analysis were collected from three independent experiments. Data were compared with Tukey's multiple comparisons test in Graphpad Prism software. Sidak's multiple comparison correction was performed.

## Glucose uptake

HmlΔ-Gal4 UAS-eGFP adults were placed on a cornmeal diet with an added 200 µL of 2-NBDG (excitation/emission maxima of ~465/540 nm, 5 mg/mL stock (used 10,000x diluted), Thermo-Fisher), which was soaked into the surface of food, immediately after infection (flies analyzed at 24 hpi) or 96 hpi (flies analyzed at 120 hpi). After 1 day, flies were prepared for microscopy (Olympus IX71). Flies for glucose uptake analysis were collected from three independent experiments.

## Activation of the hypoxia response element (HRE)

X-gal staining was performed on infected HRE-HRE-CRE-LacZ females. Flies were dipped in 75% EtOH for 1 s in order to make their cuticle non-hydrophobic and dissected in PBS. Fixation was performed with 2.5% glutaraldehyde/PBS on a LabRoller rotator for 7 min at room temperature. Adults were then washed three times in PBS. Next, two washings were performed with a PT solution (1 mL 10xPBS (Ambion), 100 µL 1M $MgCl_2 \times 6H_2O$, 300 µL 10% Triton, 8 mL $dH_2O$, 320 µL 0.1M $K_4[Fe(CN)_6]$, 320 µL 0.1 M $K_3[Fe(CN)_6]$) for 10 min. Finally, PT solution with few grains of X-gal (Sigma) was added. Samples were placed in a thermoblock at 37°C and occasionally mixed, and the colorimetric reaction was monitored. The reaction was stopped with three PBS washings at the same time for all samples. Samples for HRE activation evaluation were collected from four independent experiments.

## Concentration of metabolites

Five flies were homogenized in 200 µL of PBS and centrifuged (3 min, 4°C, 8000 RPM) for glycogen measurement. For lactate and glucose measurement, hemolymph was isolated from 25 adult males

by centrifugation (14,000 RPM, 5 min) through a silicagel filter into 50 μL PBS. Half of all samples were used for the quantification of proteins. Samples for glucose, glycogen and lactate measurement were denatured at 75°C for 10 min, whereas samples for protein quantification were stored in −80°C. Glucose was measured using a Glucose (GO) Assay (GAGO-20) Kit (Sigma) according to the manufacturer's protocol. Colorimetric reaction was measured at 540 nm. For glycogen quantification, sample was mixed with amyloglucosidase (Sigma) and incubated at 37°C for 30 min. A Bicinchoninic Acid Assay (BCA) Kit (Sigma) was used for protein quantification according to the supplier's protocol and the absorbance was measured at 595 nm. A Lactate Assay Kit (Sigma) was used for lactate concentration quantification according to the manufacturer's protocol. The absorbance was measured at 570 nm. Samples for metabolite concentration were collected from six independent experiments. Measured data were compared in Graphpad Prism using Tukey's multiple comparisons test. Sidak's multiple comparison correction was performed.

## Enzymatic activity

The enzymatic activities of lactate dehydrogenase and phosphoglucose isomerase were measured using a Lacate Dehydrogenase Activity Assay Kit (Sigma) or a Phosphoglucose Isomerase Colorimetric Assay Kit (Sigma), respectively, according to the supplier's protocol in 10,000 FACS-sorted hemocytes for each sample. Colorimetric reaction was measured at 450 nm. Samples for enzymatic activity detection were collected from six independent experiments. Measured values were compared in Graphpad Prism software using Tukey's multiple comparisons test. Sidak's multiple comparison correction was performed.

## Genotypes of experimental models

### Figure 1

(B) Hml >GFP refers to HmlΔ-Gal4 UAS-eGFP/HmlΔ-Gal4 UAS-eGFP; +/+

### Figure 2

(A–B, D–G) Hml >GFP refers to HmlΔ-Gal4 UAS-eGFP/HmlΔ-Gal4 UAS-eGFP; +/+

### Figure 3

(A) HRE-LacZ refers to HRE-HRE-CRE-LacZ/HRE HRE-CRE-LacZ; +/+
(B) Hml >GFP Ldh-mCherry corresponds to HmlΔ-Gal4 UAS-eGFP, Ldh-mCherry/HmlΔ-Gal4 UAS-eGFP, Ldh-mCherry; +/+
(C–E) Hml >GFP refers to HmlΔ-Gal4 UAS-eGFP/HmlΔ-Gal4 UAS-eGFP; +/+
(E) Hml >GFP Hif1α[RNAi] corresponds to HmlΔ-Gal4 UAS-eGFP/UAS-Hif1α[RNAi]; +/+

### Figure 4

(A, D–F) Hml x KK control corresponds to HmlΔ-Gal4 UAS-eGFP/KK control; +/+; and Hml >Hif1α[RNAi] refers to HmlΔ-Gal4 UAS-eGFP/UAS-Hif1α[RNAi]; +/+
(B) Hml >GFP refers to HmlΔ-Gal4 UAS-eGFP/HmlΔ-Gal4 UAS-eGFP; +/+; and Hml >Hif1α[RNAi] refers to HmlΔ-Gal4 UAS-eGFP/UAS- Hif1α[RNAi]; +/+
(C, G–I) Hml x TRiP control corresponds to HmlΔ-Gal4 UAS-eGFP/+; TRiP control/+; and Hml >Ldh[RNAi] refers to HmlΔ-Gal4 UAS-eGFP/+; UAS-Ldh[RNAi]/ +

### Figure 5

(A, B, C) Hml x TRiP control corresponds to HmlΔ-Gal4 UAS-eGFP/+; TRiP control/+; and Hml >Ldh[RNAi] corresponds to HmlΔ-Gal4 UAS-eGFP/+; UAS-Ldh[RNAi]/ +; and Hml >Hif1α[RNAi] refers to HmlΔ-Gal4 UAS-eGFP/UAS-Hif1α[RNAi]; +/+; and Hml x KK control corresponds to HmlΔ-Gal4 UAS-eGFP/KK control; +/+

## Figure 6

(A, C) Hml >Hif1α[RNAi] refers to HmlΔ-Gal4/+; P{tubPGal80ts}/UAS-Hif1α[RNAi]; and Hml x KK control corresponds to HmlΔ-Gal4/KK control; P{tubPGal80ts}/+

(B, D) w x Ldh[RNAi] refers to +/+; UAS-Ldh [RNAi]/+; and Hml >Ldh[RNAi] corresponds to HmlΔ-Gal4/+; P{tubPGal80ts}/UAS-Ldh[RNAi]

## Acknowledgements

The authors acknowledge funding from the Grant Agency of the Czech Republic to TD (Project 17–16406S; www.gacr.cz). JMT and GC were supported by R35 MIRA 1R35GM119557 from NIGMS/NIH. The *S. pneumoniae* bacterial strain was obtained from David Schneider. We thank to Pablo Wappner and Bruno Lemaitre, who kindly provided us with HRE-HRE-CRE-LacZ and HmlΔ-Gal4 UAS-eGFP transgenic fly lines. Other fly stocks were obtained from the Bloomington Center (Bloomington, IN) and the VDRC (Vienna, Austria). We are thankful to the reviewing editors Prof. Utpal Banerjee and Prof. Ulrich Theopold for interesting comments on our work and hints for improvement of the discussion.

## Additional information

### Funding

| Funder | Grant reference number | Author |
|---|---|---|
| Czech Science Foundation | Project 17-16406S | Tomáš Doležal |
| National Institute of General Medical Sciences | R35 MIRA 1R35GM119557 | Jason M Tennessen |

The funders had no role in study design, data collection and interpretation, or the decision to submit the work for publication.

### Author contributions

Gabriela Krejčová, Conceptualization, Data curation, Formal analysis, Investigation, Methodology, Writing—original draft, Writing—review and editing; Adéla Danielová, Pavla Nedbalová, Michalina Kazek, Lukáš Strych, Data curation, Formal analysis; Geetanjali Chawla, Jaroslava Lieskovská, Data curation; Jason M Tennessen, Resources, Data curation, Funding acquisition, Writing—original draft, Writing—review and editing; Marek Jindra, Validation, Writing—original draft, Writing—review and editing; Tomáš Doležal, Resources, Supervision, Funding acquisition, Validation, Writing—original draft, Writing—review and editing; Adam Bajgar, Conceptualization, Data curation, Formal analysis, Supervision, Validation, Investigation, Methodology, Writing—original draft, Project administration, Writing—review and editing

### Author ORCIDs

Jason M Tennessen (iD) http://orcid.org/0000-0002-3527-5683
Marek Jindra (iD) https://orcid.org/0000-0002-2196-9924
Tomáš Doležal (iD) https://orcid.org/0000-0001-5217-4465
Adam Bajgar (iD) https://orcid.org/0000-0002-9721-7534

### Decision letter and Author response

Decision letter https://doi.org/10.7554/eLife.50414.020
Author response https://doi.org/10.7554/eLife.50414.021

## Additional files

### Supplementary files

• Transparent reporting form DOI: https://doi.org/10.7554/eLife.50414.018

## Data availability

All data generated or analysed during this study are included in the manuscript and supporting files. Source data have been provided for Figures 2 and 4.

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
