## [Decision Letter]

Thank you for submitting your article "*Drosophila* macrophages switch to aerobic glycolysis to mount effective antibacterial defense" for consideration by *eLife*. Your article has been reviewed by two peer reviewers, including Utpal Banerjee as the Reviewing and Senior Editor, and Reviewer #1. The following individual involved in review of your submission has agreed to reveal their identity: Ulrich Theopold (Reviewer #2).

The reviewers have discussed the reviews with one another and drafted this decision to help you prepare a revised submission.

As you can see from the reviews, most of the comments on the general assessment of the paper can be addressed by simply rewriting some parts of the Discussion section. I have also included some experiments that I believe would strengthen the study. However following discussions with Uli Theopold, we decided that these should not be made mandatory for publication. I still include them here for your potential benefit but we believe your work is important and do not wish to hold up publication. So we leave it up to you to decide whether or not to include some of these experiments. Please do, however, attend to the suggested changes to the Discussion.

Summary:

The paper titled "*Drosophila* macrophages switch to aerobic glycolysis to mount effective antibacterial defense" by Krejčová et al. addresses the role of metabolic reprograming and activation of aerobic glycolysis in macrophages following septic infection. The authors induce sepsis in adult flies by injection of *Streptococcus* into the hemocoel and then isolate marked macrophages by FACS at different stages of the infections and monitor the changes in metabolic and transcriptional profiles during the acute and resolution stages of the macrophages and compare it with control. The principal conclusion of this study is that as in mammals, *Drosophila* macrophages also undergo metabolic reprograming and increase aerobic glycolysis, which is essential for mounting a strong defense against the infection. This process is mediated by the Hypoxia induced factor expression in normoxic conditions resulting in increased transcriptional expression of glycolytic enzymes and LDH. Additionally, the authors also describe a systemic response mediated by macrophage activation which changes the glucose metabolism based on the increase in the levels of circulating glucose and a concomitant depletion of glycogen reserves.

Essential revisions:

The experiments performed are straightforward, and, for the most part, support the conclusions reached by the authors. Both of the major conclusions – that glycolysis is activated during macrophage immune reaction, and the proposal that Hif is involved in this process – are supported by the data.

1) There is less data to describe any mitochondrial metabolic changes that occur during infection and the lack of changes in TCA cycle gene expression do not support one mechanism or other. Either attempt some of the suggested experiments, or discuss this matter as a future goal and limitation of the current study.

2) The manuscript shows that aerobic glycolysis (AG) as a means of adjusting immune reactions is conserved among invertebrates with both local and systemic effects. This might come as no surprise to *Drosophila* researchers, but evolutionary connections, particularly when it pertains to immunity is not widely appreciated and needs to be explicitly discussed.

3) There is an increasing interest in understanding how metabolism shapes immunity in mammals and establishing this in *Drosophila* is a major step forward making the phenomenon amenable to genetics. When relating the findings in this manuscript to immunology in general we would suggest you go even further and cite/discuss some recent concepts on the relevance of immune metabolism for trained immunity (for example: Netea and coworkers in Seminars in Immunol. 2016). Trained immunity has not been studied much in insects but may relate to the mechanistically poorly characterized immune priming in invertebrates.

4) What is the nature of the systemic signal? If unknown to the authors, it is still valuable to discuss its possible origins and identity.

5) Although not central to the study, mitochondrial dynamics is influenced when cells undergo metabolic reprograming to aerobic glycolysis. One of the target genes Hif-1α is PDK1, which phosphorylates PDH and renders it inactive thus blocking the TCA cycle. Increased phospho-PDH can correlate with mitochondrial fragmentation and increased ROS production. Given the interconnections between different pathways it would be important to monitor the status of the mitochondria in normal and activated macrophages in acute and resolution stages. Minimally, discuss these ideas in the manuscript.

6) In the Discussion section, the manuscript would benefit from a more in depth discussion of the evolutionary implications of immune adjustments via AG. Due to their pleiotropic character the coding regions may be under too much constraint preventing changes but genome scans for fast-evolving regions may have identified markers in regulatory regions. Is there any evidence for positive selection/sweeps for the genes in question (the authors rightly mention many are induced upon immune activation). Data may be available for both mammals and insects/invertebrates.

---

## [Author Response]

Essential revisions:The experiments performed are straightforward, and, for the most part, support the conclusions reached by the authors. Both of the major conclusions – that glycolysis is activated during macrophage immune reaction, and the proposal that Hif is involved in this process – are supported by the data.1) There is less data to describe any mitochondrial metabolic changes that occur during infection and the lack of changes in TCA cycle gene expression do not support one mechanism or other. Either attempt some of the suggested experiments, or discuss this matter as a future goal and limitation of the current study.

We agree that our paper would benefit from in depth analysis of mitochondrial dynamics. Therefore, we discuss the analysis of mitochondrial metabolism as a goal of our future study (Discussion, fifth paragraph).

2) The manuscript shows that aerobic glycolysis (AG) as a means of adjusting immune reactions is conserved among invertebrates with both local and systemic effects. This might come as no surprise to *Drosophila* researchers, but evolutionary connections, particularly when it pertains to immunity is not widely appreciated and needs to be explicitly discussed.

We are aware of the evolutionary importance of our findings and hence we discuss it (Discussion, third paragraph).

3) There is an increasing interest in understanding how metabolism shapes immunity in mammals and establishing this in *Drosophila* is a major step forward making the phenomenon amenable to genetics. When relating the findings in this manuscript to immunology in general we would suggest you go even further and cite/discuss some recent concepts on the relevance of immune metabolism for trained immunity (for example: Netea and coworkers in Seminars in Immunol. 2016). Trained immunity has not been studied much in insects but may relate to the mechanistically poorly characterized immune priming in invertebrates.

The relevance of our data for concept of trained immunity is discussed in Discussion, seventh paragraph.

4) What is the nature of the systemic signal? If unknown to the authors, it is still valuable to discuss its possible origins and identity.

In our previous work, we identified extra-cellular adenosine as an effector molecule responsible for similar effect. Nevertheless, we expect pleiotropy of several candidate molecules that we believe may serve as a systemic signal. Moreover, our current research is focused on role of gene ImpL2 in the induction of systemic metabolic changes in the context of infection (manuscript in prep.) –see Discussion, sixth paragraph.

5) Although not central to the study, mitochondrial dynamics is influenced when cells undergo metabolic reprograming to aerobic glycolysis. One of the target genes Hif-1α is PDK1, which phosphorylates PDH and renders it inactive thus blocking the TCA cycle. Increased phospho-PDH can correlate with mitochondrial fragmentation and increased ROS production. Given the interconnections between different pathways it would be important to monitor the status of the mitochondria in normal and activated macrophages in acute and resolution stages. Minimally, discuss these ideas in the manuscript.

We appreciate your ideas for our future work concerning mitochondrial metabolism. The importance of mitochondrial metabolism for the described process is obvious. We mention these ideas in the Discussion (fifth paragraph).

6) In the Discussion section, the manuscript would benefit from a more in depth discussion of the evolutionary implications of immune adjustments via AG. Due to their pleiotropic character the coding regions may be under too much constraint preventing changes but genome scans for fast-evolving regions may have identified markers in regulatory regions. Is there any evidence for positive selection/sweeps for the genes in question (the authors rightly mention many are induced upon immune activation). Data may be available for both mammals and insects/invertebrates.

Even though we tried to get deeper into the principles of population and evolutionary genetics, we are not experts in the field and we are afraid that we can interpret the data in a misleading way.

Therefore we decided just discuss aspects of evolutional conservation of hypoxia control of metabolic switch between AG and mitochondrial metabolism (Discussion, third paragraph).